# Links between Metabolic Syndrome and Hypertension: The Relationship with the Current Antidiabetic Drugs

**DOI:** 10.3390/metabo13010087

**Published:** 2023-01-05

**Authors:** Silviu Stanciu, Emilia Rusu, Daniela Miricescu, Ana Cristina Radu, Bianca Axinia, Ana Maria Vrabie, Ruxandra Ionescu, Mariana Jinga, Carmen Adella Sirbu

**Affiliations:** 1Department of Internal Medicine and Gastroenterology, Carol Davila University of Medicine and Pharmacy, Central Military Emergency University Hospital, 050474 Bucharest, Romania; 2Department of Diabetology, “Carol Davila” University of Medicine and Pharmacy, Malaxa Clinica Hospital, 02441 Bucharest, Romania; 3Department of Biochemistry, Faculty of Dental Medicine, Carol Davila University of Medicine and Pharmacy, 050474 Bucharest, Romania; 4Malaxa Clinical Hospital, 02441 Bucharest, Romania; 5Department of Cardiology, Central Military Emergency University Hospital, 050474 Bucharest, Romania; 6Department of Cardiology, Carol Davila University of Medicine and Pharmacy, Colentina Clinical Hospital, 020125 Bucharest, Romania; 7Department of Neurology, Central Military Emergency University Hospital, 050474 Bucharest, Romania

**Keywords:** hypertension, metabolic syndrome, type 2 diabetes mellitus, insulin resistance, metabolism, obesity, SGLT2-inhibitors, GLP-1 receptor agonist, tirzepatide, DPP-4 inhibitors

## Abstract

Hypertension poses a significant burden in the general population, being responsible for increasing cardiovascular morbidity and mortality, leading to adverse outcomes. Moreover, the association of hypertension with dyslipidaemia, obesity, and insulin resistance, also known as metabolic syndrome, further increases the overall cardiovascular risk of an individual. The complex pathophysiological overlap between the components of the metabolic syndrome may in part explain how novel antidiabetic drugs express pleiotropic effects. Taking into consideration that a significant proportion of patients do not achieve target blood pressure values or glucose levels, more efforts need to be undertaken to increase awareness among patients and physicians. Novel drugs, such as incretin-based therapies and renal glucose reuptake inhibitors, show promising results in decreasing cardiovascular events in patients with metabolic syndrome. The effects of sodium-glucose co-transporter-2 inhibitors are expressed at different levels, including renoprotection through glucosuria, natriuresis and decreased intraglomerular pressure, metabolic effects such as enhanced insulin sensitivity, cardiac protection through decreased myocardial oxidative stress and, to a lesser extent, decreased blood pressure values. These pleiotropic effects are also observed after treatment with glucagon-like peptide-1 receptor agonists, positively influencing the cardiovascular outcomes of patients with metabolic syndrome. The initial combination of the two classes may be the best choice in patients with type 2 diabetes mellitus and multiple cardiovascular risk factors because of their complementary mechanisms of action. In addition, the novel mineralocorticoid receptor antagonists show significant cardio-renal benefits, as well as anti-inflammatory and anti-fibrotic effects. Overall, the key to better control of hypertension in patients with metabolic syndrome is to consider targeting multiple pathogenic mechanisms, using a combination of the different therapeutic agents, as well as drastic lifestyle changes. This article will briefly summarize the association of hypertension with metabolic syndrome, as well as take into account the influence of antidiabetic drugs on blood pressure control.

## 1. Introduction

Hypertension, which possesses a significant prevalence in the general population, is one of the main constituents of metabolic syndrome. Hypertension is strongly associated with metabolic syndrome through the pathophysiology which involves obesity. Nevertheless, it represents the major risk factor responsible for elevating cardiovascular mortality and morbidity [1].

Hypertension is defined as repeated elevated office systolic blood pressure (SBP) values over 140 mmHg and/or diastolic BP (DBP) over 90 mmHg or average home BP over 135/85 mmHg [2,3].

Metabolic syndrome (MetS) has serious outcomes regarding the individual’s health, with increasing prevalence nowadays and a significant impact on healthcare systems. Its definition varied over time. MetS consists of several conditions, such as hypertension, elevated fasting glucose (over 100 mg/dL) or type 2 diabetes mellitus (T2DM), decreased high-density lipoprotein cholesterol levels (less than 40 mg/dL in men or 50 mg/dL in women), high triglycerides concentrations (over 150 mg/dL) and waist circumference over 40 inches (men) or 35 inches (women) [4].

MetS amplifies the risk for insulin resistance, cardiovascular and neurological complications [5]. The pathogenesis of MetS consists of multiple genetic and acquired mechanisms related to insulin resistance and chronic inflammation [6]. The burden of MetS is to be considered since it leads to adverse cardiovascular outcomes, which stand for the number one mortality cause worldwide [6].

Reaven, in 1988, discussed “syndrome X” as a metabolic syndrome represented by glucose intolerance, insulin resistance, dyslipidemia, hypertension and coronary artery disease [7]. However, earlier in 1973, the term “syndrome X” was first introduced by Kemp [8], representing symptoms of myocardial ischemia and electrocardiographic stress changes, but in the absence of atherosclerotic plaques of the coronary arteries [9].

Therefore, to distinguish between these two entities, Reaven’s “syndrome X” was associated with the term “metabolic” [8,9]. As we know, MetS includes central obesity, hypertension, insulin resistance and dyslipidemia. It is associated with an increased risk of developing complications such as diabetes and atherosclerotic cardiovascular disease [10]. Diabetes mellitus is a chronic metabolic disorder characterized by persistent high blood sugar levels [11]. It is classified into three types by etiology and clinical presentation, but type 2 diabetes accounts for around 90% of all cases [12]. Type 2 diabetes mellitus (T2DM) is an important problem that affects the entire planet and is constantly growing. In developed countries, diabetes is the main cause of cardiovascular diseases [12]. T2DM is characterized by insulin resistance that leads to insulin ineffectiveness. Initially, insulin production is increased to maintain glucose homeostasis, but over time it decreases Although people older than 45 years are most frequently affected, it is more and more common among young people due to multiple factors such as junk food, a sedentary lifestyle, and obesity [12]. Dysglycemia is a term used to describe impaired glucose regulation (or prediabetes) and diabetes mellitus [13]. A recent meta-analysis showed an inverse association between metabolic syndrome and intake of n-3 polyunsaturated fatty acids [14], which might explain why the increased intake of red meat decreases the incidence of dysglycemia [14]. Inadequate glucose metabolism (diabetes mellitus, impaired glucose tolerance, insulin resistance) or dyslipidemia are frequently associated with arterial hypertension [15,16,17]. In conclusion, hypertension is one of the most frequently diagnosed cardiovascular conditions, with well-known metabolic complications (hyperglycemia, diabetes mellitus, hyperlipidemia).

## 2. Pathogenesis of Hypertension in MetS

There are multiple mechanisms involved in the pathogenesis of MetS. Lifestyle, environmental, genetic, but also epigenetic [18] factors contribute to the development of MetS [6].

Insulin resistance can be found in most patients with impaired glucose tolerance or non-insulin dependent diabetes mellitus, but also in about a quarter of individuals with adequate weight and normal glucose tolerance [7]. Therefore, in these cases, the compensatory mechanism to keep glucose levels in optimal parameters is to increase insulin secretion in pancreatic cells, resulting in hyperinsulinemia [7].

Free fatty acids (FFA) play a role in the connection between insulin resistance, insulin levels and glucose tolerance. Plasma FFA can be lowered as a result of higher insulin concentrations. For that reason, hyperinsulinemia counteracts high FFA levels. When disruptions in these processes appear and hyperinsulinemia cannot be sustained, high FFA concentrations will amplify hepatic glucose production. Even slight elevations of hepatic glucose production can result in remarkable hyperglycemia [7].

There is a connection between hypertension, insulin resistance, hyperglycemia, and hyperinsulinemia [19,20,21,22,23,24,25,26,27]. Hyperinsulinemia is associated with elevated concentrations of plasmatic catecholamines, irrespective of plasmatic glucose levels [28,29]. Hypertension can also result from sodium and water metabolism through proximal renal tubular reabsorption, favoured by insulin [30].

Experimental studies indicated that hypertension could be induced in animal subjects while feeding them with food rich in fructose [7]. Due to accentuated sympathetic activity, hypertension can also be noted in sucrose-fed mice [31,32]. They conclude that blood pressure can increase as a response to dietary changes, leading to insulin resistance and hyperinsulinemia [7].

However, exercise can lower blood pressure values in people with initial hyperinsulinemia [28].

Resistance to insulin-stimulated glucose uptake is linked to hyperinsulinemia, glucose intolerance, hypertension, low HDL-cholesterol, and high triglycerides levels while also being able to influence coronary artery disease [7].

MetS is represented by a relationship between epigenetic, genetic, and environmental factors. The last two mentioned factors can lead to different expressions of orphan G protein-coupled receptors (GPCR) in MetS, such as GPR21 and GPR 82, raising the suspicion that they can represent a possible future therapeutic target gut metabolic syndrome [6,7,8,9,10,11,12,13,14,15,16,17,18,19,20,21,22,23,24,25,26,27,28,29,30,31,32,33].

MicroRNA (miRNA) are small ribonucleotide acids, which regulate gene expression at the post-transcriptional level by connecting to the 3′UTRs region of messenger RNA, being able to promote or suppress translation [34]. It was observed that suppression of miR-33a can ameliorate the functional status of MetS patients since its activity consists of favoring atherosclerosis and regulation of glucose and cholesterol metabolism. Insulin resistance, statin use, and therefore low cholesterol levels can induce SREBP 1 and 2 genes, with the overexpression of miR-33a and miR-33b. This leads to low fatty acid beta-oxidation, decreased insulin signaling and low HDL-cholesterol levels, thus raising the risk for MetS [18,19,20,21,22,23,24,25,26,27,28,29,30,31,32,33,34,35]. miR-221 and let-7 were found in high concentrations in females with MetS. Let-7 was correlated with hypertension and HDL-cholesterol levels [36].

Insulin resistance leads to hypertension through the loss of the vasodilatory effect of insulin and amplification of vasoconstriction due to FFA through reactive oxygen species [6,7,8,9,19,20,21,22,23,24,25,26,27,28,29,30,31,32,33,34,35,36,37].

Metabolomics also plays a role in the pathogenesis of MetS. Choline, L-carnitine, and trimethylamine-N-oxide are associated with insulin resistance and unfavorable cardiometabolic events [6,38,39,40,41]. Alanine, glutamine, aspartate, asparagine, arginine, histidine methionine, cysteine, and lysine play a role in the evolution of insulin resistance [42]. Early markers for the possible occurrence of MetS can be phenylalanine, tryptophan, tyrosine, and phospholipids [43,44,45].

## 3. Links between Hypertension and Metabolic Syndrome

As we know, hypertension is an important element in MetS. The connections between these two entities are multiple, and complicated and they are not fully known until now.

Links that exist between hypertension and MetS include insulin resistance, central/visceral obesity, sympathetic overactivity, activated renin-angiotensin system, oxidative stress, increased inflammatory mediators, and obstructive sleep apnea [46]. In the following lines, we will discuss each separately to see how hypertension occurs in MetS.

MetS has insulin resistance as its main constituent. Studies have shown that insulin has an anti-natriuretic effect by stimulating sodium reabsorption, an effect that is not only preserved in insulin resistance but can even be increased [46]. Therefore, this can lead to hypertension within the metabolic syndrome. Likewise, other mechanisms by which insulin resistance contributes to the appearance of elevated blood pressure in the metabolic syndrome are the loss of the vasodilator effect of insulin, vasoconstriction caused by free fatty acids, and sympathetic hyperactivation [47].

Another thing worth mentioning about the link between insulin resistance and hypertension is the observation that drugs that improve insulin resistance and reduce hyperinsulinemia (such as metformin and sensitisers glitazones) also control hypertension very well. At the same time, some antihypertensives, such as angiotensin II converting enzyme inhibitors or angiotensin II receptor antagonists, also increase insulin sensitivity [48].

The second link is visceral or central obesity. Studies have shown that adipose tissue is an endocrine organ that secretes adipocytokines such as leptin, tumour necrosis factor-α (TNF-α), interleukin-6 (IL-6), angiotensinogen, and non-esterified fatty acids (NEFA), bioactive substances that have multiple roles in the body. An important effect of adipocytokines is the production of arterial hypertension. In addition, visceral obesity is the leading cause of MetS, which is why we can establish a link between it and hypertension [46].

Regarding sympathetic overactivity, the level of serum catecholamines and the activity of the sympathetic nervous system are increased in obese patients, especially for those with central obesity. People with obesity have an active renin-angiotensin system and a positive feedback relationship with the sympathetic nervous system, contributing to high blood pressure [38]. The plasma level of leptin is increased by insulin resistance, and leptin positively influences the activity of the nervous system, which leads us to think of hypertension associated with obesity [49].

There is a connection between endothelial dysfunction, which contributes to the appearance of high blood pressure, and insulin resistance, a fact highlighted by some epidemiological studies [38]. A prospective cohort study demonstrated that endothelial vasomotor dysfunction precedes and predicts the development of hypertension [50].

Another element involved in the occurrence of elevated blood pressure is the increase of inflammatory mediators [46]. Among the mediators involved in the pathophysiology of arterial hypertension within the MetS are tumour necrosis factor-α (TNF-α) and interleukin-6 (IL-6). The serum level of TNF-α was correlated with insulin resistance and systolic blood pressure. At the same time, IL-6 stimulates the sympathetic nervous system and induces an increase in the plasmatic level of angiotensinogen and angiotensin II, causing an increase in blood pressure [46].

Sympathetic overactivity has also been associated with obstructive sleep apnea [46], another important element that can contribute to hypertension, and individuals with obstructive sleep apnea have a high prevalence of MetS [49]. In MetS, there is a sympathetic activation induced by the baroreflex dysfunction, which also characterizes obstructive sleep apnea [46]. How obstructive sleep apnea leads to sympathetic stimulation is the stimulation of arterial chemoreceptors through nocturnal hypoxia and hypercapnia [46]. Other mechanisms through which obstructive sleep apnea contributes to elevated blood pressure levels are insulin resistance, endothelial dysfunction, elevated angiotensin II and aldosterone levels, oxidative stress, hyperleptinemia, and inflammation [51,52]. In conclusion, these factors may induce vasoconstriction, decreased vasodilatation, sympathetic overactivity, and increased intravascular fluid, leading to the development of hypertension in MetS (Figure 1).

## 4. Dysmetabolic Hypertension—PROs and CONs

For many years, the constellation of hypertension, dyslipidaemia, obesity, and insulin resistance has been recognised as a distinct syndrome predisposing to the development of both type 2 diabetes mellitus and cardiovascular disease, among others [53]. On the one hand, high blood pressure, together with anthropometric and metabolic abnormalities, share common pathophysiological pathways, which partially explain why obesity and insulin resistance promotes the development of hypertension, on the other hand, how treatment of concurrent metabolic abnormalities can improve blood pressure control [54,55]. Although some controversy exists about whether MetS is indeed a syndrome or a mere association of unrelated phenotypes, the recognition of this syndrome has clinical and prognostic utility. It can help predict cardiovascular risk and guide treatment decisions, including lifestyle changes and pharmacologic treatment. The arguments in favor of the development of hypertension in patients with metabolic disturbances are controversial. Hypertension is known to be highly prevalent in patients with MetS [56]. Whether this is a causal relationship or a mere association represents a matter of discussion. Obese patients are prone to develop hypertension, but the reverse is not well established, as hypertension itself represents a poor predictor for the development of MetS [57,58]. Several mechanistic pathways that lead to a prohypertensive state have been considered in patients with MetS. Insulin resistance may promote hypertension through various processes [54]. Insulin and leptin are responsible for activating of the sympathetic nervous system, in response to over-nutrition states, which in turn increases BP values. In addition, insulin can influence the renin-angiotensin-aldosterone system, leading to an upregulation of angiotensin II type 1 receptor, potentiating peripheral vasoconstriction and plasma volume expansion [59]. These lead to a decreased response to natriuretic peptides and a rightward shift of the pressure–natriuresis curve. Furthermore, hypertensive patients with MetS are more likely to develop hypertension-mediated target organ damage than hypertensive patients alone, indicating that insulin resistance plays a central role. The components of the metabolic syndrome act synergistically to promote left ventricular hypertrophy, aortic stiffness, and microalbuminuria, which explains the increased morbidity and mortality associated with this syndrome [60].

Data from 2013 subjects from an Italian population study revealed that hypertension was present in more than 90% of individuals with MetS [61]. An increase in office blood pressure was the most frequent component found in patients with MetS, followed by hypertriglyceridemia and low plasma levels of HDL cholesterol. However, among the MetS components, only high-normal and hypertensive BP values, as well as impaired fasting glucose levels, were found to be associated with cardiovascular and all-cause mortality, while the other three components did not no prognostic association, nor did they demonstrate an increase in risk when added to the previous two risk factors. This suggests that the contribution of the different components of MetS is unbalanced, supporting the idea that metabolic syndrome is rather a heterogenous syndrome with no pathophysiological overlap [61].

On the other hand, other epidemiological data are not in accordance with this concept, suggesting that MetS is associated with an increased risk of cardiovascular events. In a study including 1750 hypertensive patients without cardiovascular disease, a significant proportion (34%) had MetS, which occurred particularly in older patients and those with longstanding hypertension. Patients with MetS had an increased risk of developing cardiovascular events than those without (hazard ratio, 3.23 vs. 1.76 per 100 patient-years; *p* value < 0.001). The association was observed even after multivariable adjustment (hazard ratio, 1.73; 95% confidence interval, 1.25 to 2.38; *p* value < 0.001), suggesting that MetS is an independent predictor of both cardiac and cerebrovascular events [62].

In another large French cohort study that included nearly 40,000 men and 20,000 women without cardiovascular disease, the prevalence of hypertension was 45% in men and 39% in women, while 19.3% of hypertensive men and 14.8% of hypertensive women had metabolic syndrome. The presence of MetS was associated with a minor increase in the risk of all-cause mortality among normotensive patients (hazard ratio, 1.09; 95% confidence interval, 0.68–1.75), while the risk among hypertensive patients was increased by 40% (hazard ratio, 1.40; 95% confidence interval, 1.13–1.74). However, no statistical significance was found after comparing the two groups. As a result, the impact of metabolic syndrome on all-cause mortality was similar between hypertensive and normotensive patients during a mean follow-up of 4.7 years. This indicates that MetS is associated with all-cause mortality, regardless of blood pressure values [63].

However, based on the existing data, we need to clearly establish whether the metabolic syndrome is indeed a clinical syndrome or a mere association of variables. The available data do not provide sufficient evidence regarding the clinical significance of MetS in hypertensive patients. Hypertension has been recognized as a major and independent risk factor for adverse cardiovascular events. It is acknowledged that hypertension tends to cluster with other metabolic abnormalities, but more efforts are needed in order to understand the pathophysiological overlap between them. This will ultimately help refine the notion of dysmetabolic hypertension.

## 5. Current Antidiabetic Drugs and the Influence of Hypertension

Globally, the prevalence of hypertension, systolic blood pressure (SBP) ≥140 mmHg, or diastolic blood pressure (DBP) ≥90 mmHg among patients with diabetes and obesity is 2–3 times higher than among those without this condition (over 60% in type 2 diabetes). In addition, a positive, linear relationship is observed between body mass index and the risk of hypertension [64].

Patients with type 2 diabetes (T2DM) and MetS frequently associate cardiovascular risk factors (CVRFs). Therefore, it has been shown that more than 70% of patients with diabetes also have hypertension or dyslipidemia.

The data provided by National Health and Nutritional Examination Survey (NHANES) indicates that 73.6% of adults with diabetes from the USA associate hypertension [65].

Apart from this, in Romania, the PREDATORR study showed that the prevalence of hypertension in diabetic patients was 61.7% [66].

Worldwide, the prevalence of dyslipidemia in diabetic patients varies between 72 and 85% [67]. Accordingly, the PREDATORR study revealed that 83.7% of patients with prediabetes and diabetes are also associated with dyslipidemia [66].

Atherogenic mechanisms operating in diabetic patients are multiple; insulin resistance and atherosclerosis are related to many factors. Novel drugs, incretin-based therapies (Glucagon-like peptide 1 (GLP-1) receptor agonist, dual-acting GLP-1 and glucose-dependent insulinotropic polypeptide (GIP) receptor agonist, dipeptidyl peptidase 4 (DPP-4) inhibitors) and renal glucose reuptake inhibitors (sodium-glucose co-transporter-2 inhibitors, SGLT2-i) through pleiotropic properties lead to cardio-reno-metabolic benefits.

Many patients with MetS do not reach the target for CVD prevention (In EUROASPIRE IV) [68]. Multifactorial therapeutic management of factors that increase cardiovascular risk, obesity, blood glucose control, treatment of hypertension, and atherogenic dyslipidemia would induce a 75% decrease in cardiovascular events.

In the Steno-2 study, which included patients with T2DM and a very high risk of cardiovascular disease, an intensive multitargeted intervention including lifestyle modification and treatment for diabetes, hypertension, dyslipidemia, and microalbuminuria, along r with secondary prevention of cardiovascular disease with antiplatelet therapy with aspirin, decreases the risk of cardiovascular and microvascular outcomes by about 50% [69].

For the first time, in 2019, joint guidelines from the European Society of Cardiology and EASD in Guidelines on Diabetes, Prediabetes and Cardiovascular Diseases recommended an SGLT-2 inhibitor (empagliflozin, canagliflozin, and dapagliflozin) or GLP-1 RAs (liraglutide, semaglutide, and dulaglutide) as first-line therapy in patients with newly T2DM with prevalent cardiovascular disease (CVD) or very high/high cardiovascular (CV) risk, such as those with target-organ damage or several CVRFs [70]. The mechanisms by which the innovative therapy decreases cardiovascular mortality and brings cardiorenal benefits can be partly explained by influencing the cardiovascular risk factors associated with type 2 diabetes: obesity, hypertension, and dyslipidemia.

Although it was initially illustrated that among all antidiabetic medications with effect on hypertension (DPP4 inhibitors, GLP-1RAs, and SGLT2-i), the most significant decrease was observed in patients treated with GLP-1 RAs [71], a recent meta-analysis indicated that SGLT2-i have a more considerable effect [72].

### 5.1. Sodium-Glucose Co-Transporter-2 (SGLT2) Inhibitors

SGLT2-i reduces the reabsorption of filtered glucose, decreases the renal threshold for glucose, and promotes urinary glucose excretion and natriuresis [73]. SGLT2 co-transporters are found in the brush border of the renal tubular cells in the first segments of the proximal tubules (S1 and S2) and are responsible for the reabsorption of 90 to 97% of filtered glucose. The blood glucose-lowering effect is intermediate/high in conditions of adequate renal function, diminishing with its deterioration. The overall cardiovascular benefits of SGLT2-i do not seem to be related to BP and cholesterol reduction [74] and may be noticed even in patients without diabetes [75].

### 5.2. Features of Hypotensive Effects of SGLT2-i

The hypotensive effect of SGLT2-i is a class effect, and the decrease is modest, with an average of 3.6/1.7 mmHg [72], obtaining a similar outcome with a small dose of hydrochlorothiazide [76].

Both systolic and diastolic values are favourably influenced. The decrease in systolic blood pressure observed in most studies was between −3.5 to −4.4 mmHg, while the reduction of diastolic blood pressure was between −1.3 to −1.9 mmHg [77].

A more significant decrease in daytime values of blood pressure (possibly related to the variability of sympathetic tone) was noticed [78].

Compared to the GLP-1RAs, the decrease in blood pressure and plasmatic volume observed in SGLT2-i is not associated with an increased ventricular frequency, which can be explained by the reduction in both cardiac preload and afterload, with maintaining the cardiac output and decreasing the sympathetic nervous system activation.

The decreasing effect of SGLT2-i on BP is ascertained immediately after administration, while the hypotensive effects of GLP-1 RAs were found only in chronic administration.

Lowering the BP is persistent and is independent of the dose of SGLT2-i, renal function, or glycemic control [77].

By primarily decreasing extracellular fluid volume, arterial hypotension is rarer than in patients treated with loop diuretics. However, in specific categories of patients (especially the elderly or those with associated or pre-existing diuretic medication), monitoring the BP more closely and, eventually, decreasing the dose of diuretic administered simultaneously is recommended.

In the EMPA-REG BP trial, treatment with empagliflozin 10 mg or 25 mg once daily for 12 weeks was associated with a significant reduction in 24-h systolic and diastolic blood pressures vs. placebo; adjusted mean (95% confidence interval) differences vs. placebo in 24-h SBP were −3.44 mmHg (95% CI: −4.78; −2.09) and −4.16 mmHg (95% CI: −5.50; −2.83), respectively; for 24-h DBP was −1.36 mmHg (95% CI: −2.15; −0.56) and −1.72 mmHg (95% CI: −2.51;−0.93), respectively [79].

In DECLARE-TIMI 58, dapagliflozin vs. placebo, at 48 months, the least squares (LS) mean ± SE was lower for SBP (132.96 ± 0.37 vs. 135.32 ± 0.37 mmHg) [80].

In a systematic review and network meta-analysis, including 424 trials (276 336 patients) compared with placebo, the weighted mean differences (WMD) for SGLT2-i on SBP levels were −2.89 mmHg (95% CI: −3.37; −2.4) [81]. The most significant placebo-subtracted reductions were evident with canagliflozin (WMD −3.38 mmHg (95% CI: −4.27; −2.49)) and empagliflozin (WMD −3.29 mmHg (95% CI: −4.16; −2.41) [66]; however, for ertugliflozin, WMD was −2.66 mmHg (95% CI: −3.86; −1.47) and for dapagliflozin −2.34 mmHg (95% CI: −3.08; −1.59) [81].

Regarding DBP, empagliflozin had the most significant lowering effect compared with placebo (WMD −1.68 mmHg (95% CI: −2.09; −1.27)) ([66]). Treatment with dapagliflozin was associated with reductions in DBP by −1.45 mmHg (95% CI: −1.87; −1.03) and canagliflozin by −1.42 mmHg (95% CI: −1.78; −1.05) [81].

Similar results were published in a previous systematic review and meta-analysis of 43 RCTs (22,528 patients) (Table 1) [77].

In another systematic review and meta-analysis, treatment with SGLT2-i was related to significant reductions in the daytime and nighttime systolic and diastolic BP [82].

Treatment with SGLT2-i can decrease BP in patients with heart failure (HF). In patients with HF, a meta-analysis including 16 RCTs, therapy with an SGLT2-i, determined a statistically significant reduction in SBP from a baseline of −1.68 mmHg (95% CI −2.70; −0.66; *p* = 0.001) vs. control [83]. Because the cardio-reno-vascular benefits are not only associated with glycemic control (Figure 2), SGLT2-i has been added to the management of HF [84,85,86].

The most significant reduction in SBP was observed for luseogliflozin, −3.42 mmHg (95% CI: −7.40; 0.56), compared to empagliflozin, −2.30 mmHg (95% CI: −3.92; −0.67), canagliflozin −1.19 mmHg (95% CI: −4.19; 1.80), and dapagliflozin −1.02 mmHg (95% CI: −3.90, 1.86), respectively [83].

In this meta-analysis of patients with HF, there was no statistically significant difference in DBP between the SGLT2-i and control groups [83].

Several studies conducted on Japanese patients with T2DM relieved significant reductions in ambulatory BP, using empagliflozin (SACRA study) [87], canagliflozin (SHIFT-J) [88], luseogliflozin (LUSCAR study) [89], and dapagliflozin (Y-AIDA study) [90].

### 5.3. Possible Mechanisms Involved in Lowering Blood Pressure Using SGLT2-i

Osmotic diuresis (110–470 mL/day) and natriuresis are responsible for the initial, early effect of decreasing BP that may be important, of −4.7 mmHg for dapagliflozin [91,92] (Figure 2).

**Figure 2 metabolites-13-00087-f002:**
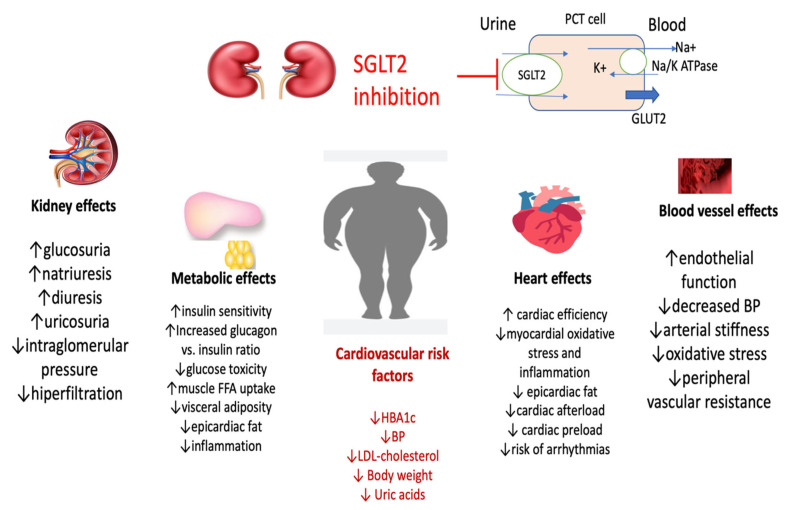
Pleiotropic effects of SGLT2-i (adapted from [91,92,95,97]) (BP, blood pressure; SGLT2-i, sodium-glucose co-transporter-2 inhibitors; PCT, proximal convoluted tubule; GLUT2, Glucose transporter 2, K, potassium; Na, sodium, ATP, adenosine triphoshat; FFA, fatty free acids).

Decreasing the sympathetic nervous system activity may be responsible for the persistence of the hypotensive effect of SGLT2-i (Figure 2) [93].

For decreasing of body weight, it should be mentioned that, in some studies, the effect of lowering the BP was independent of the weight loss. The glucosuric effect of SGLT2-i is associated with weight loss (Figure 2). In a recent meta-analysis, treatment with SGLT2-i, compared with placebo, generated a significant mean weight reduction of −1.79 kg (95% CI: −1.93; −1.66, *p* < 0.001) [94]. In patients with HF, SGLT2-i reduced body weight by −1.36 kg; (95% CI: −1.68; −1.03; *p* < 0.001) [83].

SGLT2-i suppresses the renal renin-angiotensin system by increasing the sodium concentration at the level of the juxtaglomerular apparatus [95] and decreasing of vascular rigidity by direct effect [96] or anti-inflammatory properties (Figure 2) [97].

Other mechanisms suggested to be involved in lowering BP are oxidative stress and enhancement of endothelial dysfunction (Figure 2) [98,99]. These combined actions result in a significant decrease in BP.

The effect of SGLT2-i on blood pressure remains unchanged regardless of the dose of SGLT2-i [99], independent of renal function and glycemic control [100,101].

Side effects of SGLT2-i include mycotic genital infections (vaginitis in women, balanitis in men), urinary infections, volume depletion, arterial hypotension, and dizziness (especially in the elderly or in combination with diuretics, angiotensin-converting enzyme inhibitors or angiotensin II receptor blockers), Fournier’s gangrene of the genital organs; distal lower limb amputations and fractures (for canagliflozin) [70,74,77,80].

A rare complication, especially in patients with severely impaired insulin secretory function, is diabetic ketoacidosis [70,74,77,80].

The kidney, metabolic, and cardiovascular effects explain possible mechanisms involved in lowering blood pressure using SGLT2-i. Kidney effects include the rise of glucosuria, natriuresis, uricosuria and diuresis, reduced intraglomerular pressure, and hyperfiltration.

Metabolic effects can be explained by increasing insulin secretion and glucagon to insulin ratio and reducing glucose toxicity; on lipids, metabolism reducing visceral adiposity, epicardiac fat and inflammation; consecutive increased muscle FFA uptake. Heart effects, incompletely elucidated, include reduced cardiac pre- and afterload, but also improvement in cardiac efficacity, reduced myocardial oxidative stress and inflammation consecutive with decreased epicardiac fat.

At the level of the blood vessel, improved endothelial function, decreased oxidative stress, arterial stiffness and peripheral vascular resistance.

## 6. GLP-1 Receptor Agonist (GLP-1RAs)

GLP-1 (glucagon-like peptide-1) is an incretin hormone produced by differential posttranslational processing of the proglucagon protein by the enteroendocrine L cells” [102].

GLP-1 receptor agonists are drugs administered as subcutaneous injections (the exception is semaglutide, for which there are also formulations for oral administration). These can be with short-acting (exenatide twice daily, lixisenatide once daily, and oral semaglutide once daily) and long-acting agents (liraglutide—once daily; albiglutide, dulaglutide, exenatide, and semaglutide administrated once weekly). The glycemic effects of GLP-1 are mainly mediated by binding to its selective heptahelical G-protein–coupled receptor GLP-1R and the formation of cAMP via Gs signaling [103].

Glucose control is attained through several mechanisms of action: augmentation of glucose-dependent insulin secretion, suppressed glucagon secretion, reduced appetite, slowed gastric emptying, and concomitant reduction of food intake.

Moreover, GLP-1RAs also exert beneficial roles in multiple organ systems in which the GLP-1 receptors exist, including the cardiovascular system. In clinical trials, the identified effect of GLP-1RAs on BP differed from a slight increase, as suggested in a study with dulaglutide ([104]), to neutrality or a substantial BP-lowering impact [105,106,107].

### 6.1. BP Changes Observed in Clinical Trials with GLP-1 RAs

The administration of GLP-1 in intravenous infusion does not cause a decrease in BP [108]. Contrariwise, in a small study, including ten overweight men, intravenous administration of exenatide acutely increases heart rate (HR) by 6.8 beats/min (95% CI: 1.7; 11.9, *p* < 0.05) and increases SBP by 9.8 mmHg (95% CI: 3.5;16.1, *p* < 0.01) [109].

In chronic administration, a moderate decrease in SBP was observed [110]. In a meta-analysis including 32 trials, GLP-1 agonists decreased systolic blood pressure by −1.79 mmHg (95% CI: −2.94; −0.64) and −2.39 mmHg (95% CI: −3.35; −1.42) compared to placebo and active control, respectively ([93]). There was no statistically significant reduction in diastolic blood pressure −0.54 mmHg (95% CI: −1.15; 0.07) vs. placebo and −0.50 mmHg (95% CI: −1.24; 0.24) vs. active control) [111].

The most important BBP lowering effect was found with semaglutide 1 mg administered weekly vs. exenatide ten mcg with daily administration (−4.6 mmHg vs. −2.2 mmHg) without significant difference in DBP [112].

Dulaglutide 1.5 mg significantly reduced mean 24-h SBP at 16 and 26 weeks vs. placebo (least squares mean difference −2.8 mmHg (95% CI: −4.6; −1, *p* ≤ 0.001) at 16 wk and −2.7 mmHg (95% CI: −4.5; −0.8, *p* ≤ 0.002 at 26 wk ([95]). Dulaglutide 1.5 mg increase heart rate by 2.8 bpm (95% CI: 1.5, 4.2) [113].

Overall, GLP-1 agonists increased the HR by 1.86 beats/min (bpm) (95% CI: 0.85; 2.87) vs. placebo and 1.90 bpm (95% CI: 1.30; 2.50) vs. active control [111,114].

Simultaneously with the decrease in BP values, an increase in ventricular frequency was also reported [110]. It may be explained by the enhancement of the sympathetic nervous system (SNS) [115] related to the inhibition of the autonomic nervous system, especially the parasympathetic nervous system [116], with a reduced total peripheral resistance [117]. The sinoatrial node stimulation contributes to GLP-1RA–related increases in heart rate [116].

In a systematic review and network meta-analysis, including 424 trials (276 336 patients) compared with placebo, the weighted mean differences (WMD) for GLP-1RAs on SBP levels varied from 2.93 to 2.34 mmHg [81]—with exenatide ER and dulaglutide being the least efficient (Table 2) [81]. In terms of DBP, decline ranged from 1.02 mmHg for exenatide twice-daily to −0.53 mmHg for semaglutide PO; dulaglutide, exenatide ER, and lixisenatide had a neutral effect [81].

### 6.2. Mechanisms Possibly Involved in Influencing BP Values

Data from several multicenter, long-term cardiovascular outcome trials (CVOTs) with GLP-1 RAs indicated cardiovascular benefit in patients with T2DM with CVD or at very high/high risk (Lixisenatide ELIXA [118], Liraglutide LEADER [119], Semaglutide SUSTAIN 6 [120], Exenatide EXSCEL [121], Albiglutide Harmony Outcomes [122], Dulaglutide REWIND [123], and Oral semaglutide PIONEER 6 [124].

In this study, treatment with GLP-1 RAs revealed a slight but significantly reduced BP. The cardiovascular effects of GLP-1 RAs might be facilitated by improving cardiovascular risk factors such as HBA1c, BP, and body weight, reducing anti-inflammatory markers (hs-CRP) (Figure 3). In addition, it decreases vascular resistance partially by increasing nitric oxide production at the endothelium level [125] and reducing endothelial dysfunction by inhibiting oxidative stress and inflammation [126] (Figure 3).

GLP-1 RAs act at the level of the myocardium through specific receptors and inhibit cardiomyocyte apoptosis [127]. In experimental models, GLP-1 improves myocardial function and cardiac output, improves insulin sensitivity, and myocardial glucose utilisation. In animal models of myocardial ischemia, GLP-1 administration reduced infarct size, and in human subjects with acute myocardial infarction and severe systolic dysfunction after successful primary angioplasty, GLP-1 administration improved regional and global left ventricular function [128].

GLP-1 RAs increase natriuresis and diuresis reducing BP [129], in part, to inhibition of the proximal tubule Na+/H+ exchanger isoform 3 in the proximal renal tubule [130,131].

At the CNS level, GLP-1 RAs increase sympathetic activity and decrease vagal activity [116].

The renin-angiotensin-aldosterone system (RAAS) is modulated at the renal level [132]. In twelve healthy male subjects, synthetic human GLP-1 infusion decreased the circulating concentration of ANG II by 19% (95% CI: −28; −8%, *p* = 0.003) without significant changes for renin or aldosterone or other components of the renin-angiotensin-aldosterone system [132]. Liraglutide decreased angiotensin II (ANG II) level by 21% (*p* = 0.02), but there were no effects on other renin-angiotensin system components, atrial natriuretic peptides (ANPs), metanephrine or excretion of catecholamines [133].

Dulaglutide was not associated with significant changes in serum aldosterone, plasma renin activity, plasma metanephrines, normetanephrine, or N-terminal pro-brain natriuretic peptide [113].

In addition, GLP1-RAs may improve endothelial cell function, have anti-proliferative effects on smooth muscle cells, limit activation and recruitment of macrophages in atherosclerotic plaques, decrease inflammatory cytokines, and increase endogenous antioxidant defences [134].

Weight loss is important for treating hypertension, being associated with BP reduction [135]. In a study with dulaglutide, no correlation was found between weight loss and BP reduction [113]. The most common side effects of GLP-1RAs are nausea, vomiting, and diarrhoea, which decrease over time [136,137,138]. Data from clinical studies and medical practice attest that GLP-1 Ras do not predispose to acute pancreatitis, pancreatic adenocarcinoma or medullary thyroid carcinoma [136,137,138].

At the kidney level, GLP-1 RAs increase natriuresis and diuresis, lowering BP and decreasing renal inflammation and oxidative stress. On the blood vessel, decrease vascular resistance partially by raising nitric oxide production at the endothelium level and reducing endothelial dysfunction by inhibiting oxidative stress and inflammation. GLP-1 RAs at the level of the myocardium inhibit cardiomyocyte apoptosis, improve myocardial function and cardiac output, improve myocardial glucose utilization, and reduce inflammation.

**Figure 3 metabolites-13-00087-f003:**
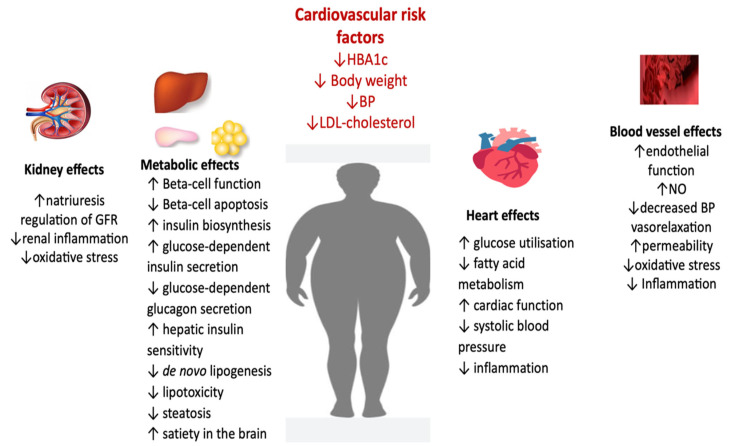
Pleiotropic effects of GLP-1 RA (adapted from [136,137,138]) (BP, blood pressure; FFA, fatty free acids; LDL, low-density lipoprotein; GFR, glomerular filtration rate; NO, nitric oxide).

The combination of GLP-1 RAs and SGLT2-is is of interest because of complementary mechanisms of action: GLP-1 RAs enrich insulin secretion, slow gastric emptying, and lower body weight, and SGLT2-i facilitate urinary glucose excretion and decrease body weight. Agents from both classes have been demonstrated to reduce CV risk. Thus, combined treatment including a GLP-1 RAs and an SGLT2-1, with or without metformin, should be the best choice to start therapy for T2DM with CVRFs (obesity, hypertension, dyslipidemia) because, in addition to reducing glucose levels, body fat will also be decreased as well BP and cardiovascular risk. With these modifications, there is the possibility for a decline in cardiac outcomes, cardiac and total mortality, and a slowdown of the decrease in renal function.

### 6.3. Tirzepatide, a Dual GIP (Glucose-Dependent Insulinotropic Polypeptide)/GLP-1 (Glucagon-like Peptide-1) Receptor Co-Agonist

In the SURMOUNT-1 trial [139], a significant decrease in 24 h ambulatory BP was found, with a potential beneficial effect in reducing cardiovascular risk in overweight or obese people. The reduction in blood pressure values was associated with an insignificant increase in ventricular frequency.

Thus, SBP decreased from the initial value by 5.6, 8.8, and 6.2 mmHg in patients who received doses of 5, 10, and 15 mg of tirzepatide weekly, while in the control group, SBP increased by 1.8 mmHg. DBP values in the three treatment groups decreased on average by 1.5, 2.4, and 0.0 mmHg, while they increased by 0.5 mmHg in the placebo-controlled group. This clinically significant BP reduction observed in the first 24 weeks of treatment is superior to the BP-lowering effect observed in studies with GLP-1 Ras [139].

The ventricular rate decreased on average by 1.8 bpm in the control group and increased by 0.3, 0.5, and 3.6 bpm, respectively, in the group treated with tirzepatide, similar to the effect of GLP-1 Ras [139].

In this study, the curve of decrease in blood pressure values was steep in the first 24 weeks of treatment, so, later, it remained on the plateau.

It should be mentioned, however, that the blood pressure values of the patients enrolled in the SURMOUNT-1 study were normal, one of the exclusion criteria being a BP value >160 mmHg [139].

The hypotensive effect is expected to be more consistent in hypertensive patients.

By acting on cardiovascular risk factors (significant weight loss, lowering BP, and improving the metabolic profile through the glycemic and lipid components), tirzepatide can reduce cardiovascular risk [139].

### 6.4. DPP-4 Inhibitors

Few studies have analyzed the effects of DPP-4 inhibitors on BP in T2D. Some studies indicate that sitagliptin may decrease SBP [140,141,142], although this result has not been observed in all studies [143]. Linagliptin, 5 mg once daily vs. placebo, was associated with small decreases in SBP and DBP [144]. Acute administration of vildagliptin reduced SBP and DBP and raised HR without affecting superior mesenteric artery blood flow [145]. In chronic administration, vildagliptin is associated with a decrease in SBP (−2.6 ± 0.3, *p* < 0.001) and DBP (−1.64 ± 0.8, *p* < 0.001) [146].

## 7. New Antihypertensive Drugs, New Perspectives

It is important to attain and maintain an optimal BP target (130/80 mmHg) ([129]) to reduce the risk of the progression of chronic micro- and macrovascular complications. Individualization of treatment targets is based on clinical characteristics (high or very high cardiovascular risk, chronic kidney diseases (CKD), stroke, older age, frailty) and potential adverse effects of antihypertensive treatment (syncope, orthostatic hypotension, kidney injury) [147].

Treatment for hypertension in patients with diabetes should include any of the antihypertensive pharmacotherapy drug classes with demonstrated to reduce cardiovascular risk: angiotensin-converting (ACE) inhibitors, angiotensin receptor blockers (ARB), thiazide-like diuretics (chlorthalidone and indapamide), or dihydropyridine calcium channel antagonists, and the mineralocorticoid receptor antagonists finerenonă [147,148].

Nontraditional BP-lowering agents such as SGLT2-i and GLP-1 ARs can be used, but monotherapy may be inadequate to control BP [148].

In patients with resistant hypertension, the addition of a mineralocorticoid receptor antagonist (MRA) may be considered. Recent studies assume that sacubitril/valsartan could be used in the treatment of patients with resistant hypertension, with or without additional MRA therapy [149].

### Nonsteroidal Mineralocorticoid Receptor Antagonist

Finerenone is a new, selective, nonsteroidal MR antagonist with a more selective activity than spironolactone and eplerenone. Finerenone blocks MR-mediated sodium reabsorption and mineralocorticoid receptor overactivation [150]. The benefits are related to anti-inflammatory and anti-fibrotic effects observed in preclinical models [150].

Three extensive studies have been published on finerenone: FIDELIO-DKD [151], FIGARO-DKD [152], and FIDELITY [153].

In FIDELIO-DKD, finerenone showed a significant reduction in the primary kidney composite outcome, lowering the risk for CKD progression (hazard ratio, 0.82; 95% CI 0.73; 0.93, *p* = 0.001) among patients with predominantly stage 3–4 CKD with severely increased albuminuria and T2DM; a lower risk of cardiovascular event was observed in the finerenone group [151].

In FIGARO-DKD study, finerenone significantly improved cardiovascular composite outcomes vs. placebo, in patients with T2DM and CKD (moderately elevated albuminuria with stage 2–4 CKD or severely elevated albuminuria and stage 1–2 CKD) [152].

The FIDELIO-DKD and FIGARO-DKD trials show that finerenone has a modest impact on SBP in patients with DKD. The mean SBP decline was −2.1 mmHg at 12 months in FIDELIO-DKD [133], and −2.85 mmHg at 12 months in FIGARO-DKD [134]. In FIDELITY, the benefit on SBP was modest; in the finerenone group, was noticed the decrease of mean SBP with −3.2 mmHg at four months, −2.5 mmHg at 12 months, and −1.8 mmHg at 48 months [153].

Finerenone is recommended for patients with T2DM, an eGFR 25 mL/min/1.73 m^2^, normal serum potassium, and urinary albumin-to-creatinine ratio >30 mg/dL in addition to an ACE inhibitor an ARB at the maximum tolerated dose [130] or for persistent albuminuria in addition to a renin-angiotensin system inhibitor and SGLT2-i or in people with T2DM and CKD who cannot take an SGLT2-i [148].

In the FIDELITY study, at baseline, some of the included patients received treatment with SGLT2-i (6.7%) or GLP-1 RAs (7.2%). The cardiorenal benefits are maintained regardless of whether there is an SGLT2-i or GLP-1 RAs in the treatment [153].

Esaxerenone, another novel non-steroidal MRA, was authorised to treat hypertension and diabetic CKD [154].

In monotherapy, esaxerenone was associated with decreased BP during the study period −18.5/−8.8 mmHg; add-on to a RAS inhibitor, and the decline was −17.8/−8.1 mmHg [119].

In the ESAX-DN study, in patients with T2DM with microalbuminuria, esaxerenone showed a raised probability of normalising a higher urinary albumin-to-creatinine ratio and declining the progression of albuminuria [155].

In conclusion, there is much evidence from several CVOTs that indicate cardio-reno-metabolic benefits using an SGLT2-i and GLP-1 RAs in patients with metabolic syndrome at very high/high CV risk or with atherosclerotic cardiovascular diseases. SGLT2-i (empagliflozin, canagliflozin, dapagliflozin) yielded a more marked BP decline (SBP/DBP –2.46/–1.46 mmHg) without heart rate differences [70]. The BP-lowering effects of these drugs must be considered when managing BP. Treatment with these innovative molecules should be started as early as possible in type 2 diabetes. Furthermore, finerenone decreases the risk of cardiovascular outcomes and kidney disease progression in a broad spectrum of patients with CKD and T2DM [153].

## 8. Conclusions

Given that MetS is a constellation of abnormalities such as abdominal obesity, dyslipidemia, hypertension, and hyperglycemia, its treatment includes medication capable of targeting each of these elements. Thus, metabolic diseases are treated with anti-obesity, lipid-lowering, anti-hypertensive and anti-diabetic drugs.

Firstly, lifestyle changes are very important in patients with MetS. Losing weight can increase insulin sensitivity, reducing the risk of type 2 diabetes and can lower blood pressure. This can be achieved through diet and regular exercise. However, a potential solution for treating metabolic diseases would be the development of drugs with multiple actions. An ideal drug for MetS therapy involves decreased weight, blood pressure, inflammation, plasma lipids, and blood glucose levels.

Secondly, better adherence, rethinking the initial antihypertensive pharmacotherapy, using the triple or quadruple fixed combination that includes small doses of different therapeutic agents, thus targeting multiple mechanisms involved in the pathogenesis of hypertension, is also important.

The data of the latest studies have led to a paradigm shift regarding the management of cardiometabolic disorders. The patient-centred approach and new therapeutic classes improve the glycemic balance and reduce numerous cardiovascular risk factors. Early identification and treatment of cardiometabolic factors and conditions associated with metabolic syndrome will be related to a favourable impact on mortality and morbidity. The last years’ research brings hope in obtaining better blood pressure control, in parallel with cardio-reno-metabolic protection.

## Figures and Tables

**Figure 1 metabolites-13-00087-f001:**
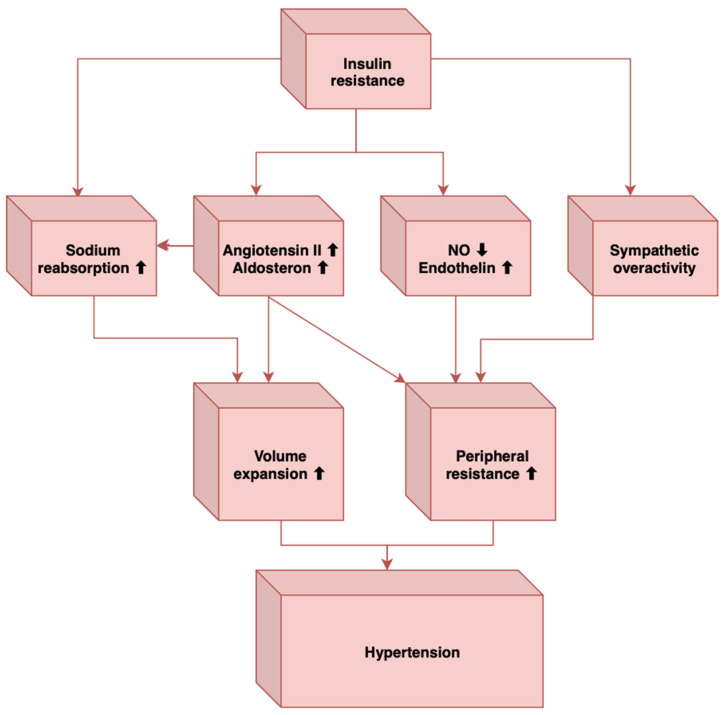
The main mechanisms that link insulin resistance to hypertension (Adapted from [51]).

**Table 1 metabolites-13-00087-t001:** Effects of SGLT2-i on blood pressure [77].

		WMD	Upper CI	Lower CI
SBP	SGLT2-i	−2.46	−2.86	−2.06
	Canagliflozin	−2.23	−2.28	−2.18
	Dapagliflozin	−1.03	−1.09	−0.97
	Empagliflozin	−2.59	−2.7	−2.49
		WMD	Upper CI	Lower CI
DBP	SGLT2-i	−1.46	−1.82	−1.09
	Canagliflozin	−2.23	−2.3	−2.16
	Dapagliflozin	−0.72	−0.78	−0.66
	Empagliflozin	−1.09	−1.18	−1.01

CI, confidence interval; DBP, diastolic blood pressure; SBP, systolic blood pressure; SGLT2-i, sodium-glucose co-transporter-2 inhibitors; WMD, weighted mean difference.

**Table 2 metabolites-13-00087-t002:** Effects of GLP-1RAs on blood pressure (adapted after [81]).

		WMD	Upper CI	Lower CI
SBP	GLP-1 RAs	−2.33	−2.86	−1.80
	Semaglutide PO	−3.06	−4.21	−1.91
	Semaglutide SC	−2.93	−3.98	−1.90
	Exenatide	−2.68	−4.03	−1.34
	Liraglutide	−2.48	−3.24	−1.73
	Exenatide ER	−1.76	−2.82	−0.70
	Dulaglutide	−1.34	−2.36	−0.31
	Lixisenatide	−0.4	−2.32	−1.52
DBP	GLP-1 RAs	WMD	Upper CI	Lower CI
	Exenatide	−1.03	−1.73	−0.33
	Lixisenatide	−0.96	−2.23	−0.31
	Semaglutide SC	−0.68	−1.17	−0.20
	Semaglutide PO	−0.53	−1.06	0.00
	Liraglutide	−0.17	−0.55	−0.21
	Exenatide ER	−0.13	−0.64	−0.39
	Dulaglutide	0.24	−0.20	0.68

CI, confidence interval; DBP, diastolic blood pressure; GLP-1 RAs, Glucagon-like peptide 1 (GLP-1) receptor agonist SBP, systolic blood pressure; WMD, weighted mean difference.

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
