# Peer review of "Links between Metabolic Syndrome and Hypertension: The Relationship with the Current Antidiabetic Drugs"

_metabolites, 2023, doi:10.3390/metabo13010087_

Round 1

Reviewer 1 Report

My compliments with the authors for the review, but there are some english mistakes through the text that must me corrected. The authors describe the metabolic syndrome but not the type 2 diabetes. It is well known that for the metabolic syndrome the only treatment authorized for weight loss is Liraglutide in 3 mg/die e not the SGLT2i or the other GLP!RAS or DPPIV (although there are on going studies regarding GLP1RA and their use in obesity). For that, authors must included the definition of type 2 diabetes and is useful to add the reference "the molecular link between oxidative stress and type 2 diabetes: a target for new therapies against cardiovascular diseases". Then the author they describing the anti hypertension effect and the importance of the SGLT2i in the hypertension, but is lack of data, it will be useful to add " the new role of SGLT2 inhibitors in the management of heart failure: current evidence and future perspective" so thet can underline the importance also in heart failure.

Author Response

On behalf of our research group, I would like to thank you for your time and your comments. We highly appreciated your recommendations and taking into consideration your suggestions, we have made the following changes to our manuscript:

1.My compliments with the authors for the review, but there are some english mistakes through the text that must me corrected. Thank you for this observation, we have corrected the English text.

  1. The authors describe the metabolic syndrome but not the type 2 diabetes. It is well known that for the metabolic syndrome the only treatment authorized for weight loss is Liraglutide in 3 mg/die e not the SGLT2i or the other GLP!RAS or DPPIV (although there are ongoing studies regarding GLP1RA and their use in obesity). For that, authors must included the definition of type 2 diabetes and is useful to add the reference "the molecular link between oxidative stress and type 2 diabetes: a target for new therapies against cardiovascular diseases".

Thank you for this important remark, we added to the text the recommended changes, lines 78-100.

Also, we added 2 references for the mentioned phrase (54, 55), line 258

3.Then the author they describing the anti hypertension effect and the importance of the SGLT2i in the hypertension, but is lack of data, it will be useful to add " the new role of SGLT2 inhibitors in the management of heart failure: current evidence and future perspective" so thet can underline the importance also in heart failure.

Thank you for this remark, we have introduced a phrase  regarding this aspect, lines 429-431

In conclusion, we have uploaded on the platform the new version of our manuscript according to the recommendations and hope to fulfill your requirements and make it easier to read. Thank you for taking into consideration the publishing of our manuscript.  

Kind regards,

Dr. Emilia Rusu

[email protected]

Dr. Daniela Miricescu

[email protected]

Reviewer 2 Report

The work presented is of particular interest to the pharmacological area. Although there are some works similar to the one presented here, the manuscript has relevance for a wide range of readers. 

However, there are some points to consider before publication.

1. The title does not reflect the manuscript, I believe that the concept of metabolic syndrome should be included in the title. 

2. There are complete paragraphs without citations or very few, for example on page 6 and in section 6.4.

3. A review article should consider the latest publications, however, this manuscript has a large number of references more than 5 years old. 

4. Review the references in detail, the manuscript has some errors, for example references 64 and 128.

5. Figure 2 is not mentioned in the text.

6. I consider that figures 2 and 3 should be explained in the figure captions and not only put the meaning of the acronyms. 

7. Check that all references have been cited in the text, reference 88 is not observed in the manuscript.

8. Review the manuscript, as there are some typographical errors, for example in line 648. 

Author Response

On behalf of our research group, I would like to thank you for your time and your comments. We highly appreciated your recommendations and taking into consideration your suggestions, we have made the following changes to our manuscript.

1.The title does not reflect the manuscript, I believe that the concept of metabolic syndrome should be included in the title. 

Thank you for this  remark, we have changed the title, lines 2-4

  1. There are complete paragraphs without citations or very few, for example on page 6 and in section 6.4.

Thank you for this remark, we have added references to the text , sections 4 and 6.4, lines 251-273, and 622-634                                                                                                                                              

  1. A review article should consider the latest publications, however, this manuscript has a large number of references more than 5 years old. 

Thank you for this remark, the old references are very important for the topic because they were the first who described the phenomena. We also added references less than 5 years, references, 10, 12, 13, 14, 54, 56, 59, 84, 85, 86

  1. Review the references in detail, the manuscript has some errors.

Thank you for this remark, we corrected the references.

  1. Figure 2 is not mentioned in the text.

Thank you for this remark,  Figure 2 has been added to the text, lines 446 and 451

  1. I consider that figures 2 and 3 should be explained in the figure captions and not only put the meaning of the acronyms. 

Thank you for this remark, we have added additional explanations, lines 475-485 for figure2, and lines 593-599 for figure 3

  1. Check that all references have been cited in the text, reference 88 is not observed in the manuscript.

Thank you for this remark,  we added 88 reference to the text, line 506

  1. Review the manuscript, as there are some typographical errors, for example in line 648.

Thank you for this remark,  we rewrote the paragraphs, lines 110-113, 126-130, 193-195, 691-697

In conclusion, we have uploaded on the platform the new version of our manuscript according to the recommendations and hope to fulfil your requirements and making it easier to read. Thank you for taking into consideration the publishing of our manuscript.  

Kind regards,

Dr. Emilia Rusu

[email protected]

Dr. Daniela Miricescu

[email protected]

Reviewer 3 Report

1.       A little review is lacking regarding the side effects of the drugs and the intertherapeutic effects. The conclusions do not differentiate hypertension from all drug treatment for MetS because weight and hypertension are very closely related and finally the conclusion from a study that started with a focus on blood pressure is treatment for all MetS.

2.       It is possible that choosing a sample of people with high blood pressure who are not obese, or diabetic could have been more focused on the research topic.

3.       In table 2, a row of GLP1 measurement in DBP is missing. There is a GLP10 measurement only in SBP in the first line.

4.       Line 648 begins the section with the sentence "or better" which should be part of a previous idea and cannot be the beginning of a sentence.

Author Response

On behalf of our research group, I would like to thank you for your time and your comments. We highly appreciated your recommendations and taking into consideration your suggestions, we have made the following changes to our manuscript:

  1. A little review is lacking regarding the side effects of the drugs and the intertherapeutic effects. The conclusions do not differentiate hypertension from all drug treatment for MetS because weight and hypertension are very closely related and finally the conclusion from a study that started with a focus on blood pressure is treatment for all MetS.

Thank you for this remark, we added a paragraph regarding these aspects, lines 463-469 (for SGLT-i) and lines 463-469 (for GLP-1RAs). We also performed modifications regarding the conclusion section, lines 710-732

  1. It is possible that choosing a sample of people with high blood pressure who are not obese, or diabetic could have been more focused on the research topic.

Thank you for this remark, but this idea will be the subject of another article.

  1. In table 2, a row of GLP1 measurement in DBP is missing. There is a GLP10 measurement only in SBP in the first line.

Thank you for this remark, we have modified

  1. Line 648 begins the section with the sentence "or better" which should be part of a previous idea and cannot be the beginning of a sentence.

Thank you for this remark, we corrected it, line 721

In conclusion, we have uploaded on the platform the new version of our manuscript according to the recommendations and hope to fulfil your requirements and making it easier to read. Thank you for taking into consideration the publishing of our manuscript.  

Kind regards,

Dr. Emilia Rusu

[email protected]

Dr. Daniela Miricescu

[email protected]